# A Real-Time Apple Targets Detection Method for Picking Robot Based on ShufflenetV2-YOLOX

**Wei Ji \*, Yu Pan, Bo Xu and Juncheng Wang**

School of Electrical and Information Engineering, Jiangsu University, Zhenjiang 212013, China;
ailqy369@163.com (Y.P.); xubo@ujs.edu.cn (B.X.); w18981051316@icloud.com (J.W.)
\* Correspondence: jiwei@ujs.edu.cn

**Abstract:** In order to enable the picking robot to detect and locate apples quickly and accurately in the orchard natural environment, we propose an apple object detection method based on Shufflenetv2-YOLOX. This method takes YOLOX-Tiny as the baseline and uses the lightweight network Shufflenetv2 added with the convolutional block attention module (CBAM) as the backbone. An adaptive spatial feature fusion (ASFF) module is added to the PANet network to improve the detection accuracy, and only two extraction layers are used to simplify the network structure. The average precision (AP), precision, recall, and F1 of the trained network under the verification set are 96.76%, 95.62%, 93.75%, and 0.95, respectively, and the detection speed reaches 65 frames per second (FPS). The test results show that the AP value of Shufflenetv2-YOLOX is increased by 6.24% compared with YOLOX-Tiny, and the detection speed is increased by 18%. At the same time, it has a better detection effect and speed than the advanced lightweight networks YOLOv5-s, Efficientdet-d0, YOLOv4-Tiny, and Mobilenet-YOLOv4-Lite. Meanwhile, the half-precision floating-point (FP16) accuracy model on the embedded device Jetson Nano with TensorRT acceleration can reach 26.3 FPS. This method can provide an effective solution for the vision system of the apple picking robot.

**Keywords:** machine vision; picking robot; apple detection; YOLOX; ShufflenetV2



## 1. Introduction

China's apple planting area and output account for more than 50% of the world [1], but its picking is still dominated by manual picking, with high cost. Therefore, the apple picking robot is the development direction in the future. How to locate and detect apples quickly and accurately in the natural environment is the focus and difficulty of vision research of picking robots [2].

At present, the research on fruit detection at home and abroad is mainly divided into target detection based on the traditional algorithm and target detection based on the deep learning algorithm, and both have made some progress. Traditional algorithms require artificially designed features [3], and their accuracy and detection speed are not as good as those of deep learning algorithms. Currently, they are mostly used for image preprocessing. Xia [4] proposed a method for fruit segmentation based on the K-means clustering algorithm. The Canny edge detection operator was used to extract the fruit contour, the Y-node search algorithm was used for contour separation, and finally, the least squares method was used for contour reconstruction. Liu [5] used a simple linear iterative clustering algorithm to segment the apple image collected in the orchard into super-pixel blocks, and used the color features extracted by blocks to determine the target candidate region. Lv [6] calculated the distance of each fruit in the connected area by using the Euclidean distance method, extracted the effective peak from the smoothed curve by using the improved local extreme value method, and determined the shape of overlapping apples according to the number of peaks. Bochkovskiy [7] chose incandescent lighting to obtain images at night. In the image segmentation stage, the power transformation was

used to improve the R-G color difference threshold segmentation method, and the genetic algorithm was introduced to optimize the solution of the maximum interclass variance. The accuracy was 94% and the detection speed was 2.21 FPS.

The detection algorithm based on deep learning has wider applicability than the traditional algorithm. When using a specific dataset, it can learn deeper features and obtain higher accuracy. It is easier to detect the target. In recent years, deep learning has been used in a wide range of industries. Some scholars have conducted in-depth research on apple target detection based on deep learning. Sa [8] basically achieved rapid detection and achieved an F1 score of 0.838 by using the improved Fast R-CNN training RGB color and near-infrared images to detect fruits. Zhao [9] used the improved YOLOv3 algorithm with 13 layers to prove that it is feasible to use the deep learning algorithm in the natural environment under the verification of different illumination directions, different growth stages of apples, and different picking times. Mazzia [10] achieved a detection speed of 30 FPS using a modified YOLOv3-Tiny network on a matched embedded device, the Jetson AGX Xaver. However, the Jetson AGX Xaver is very expensive and its AP is only 83.64%, which does not satisfy the need for detection accuracy. Yan [11] using the improved YOLOv5 can effectively identify grasping apples that are not obscured by leaves or only obscured by leaves, and nongrasping apples that are obscured by branches or other fruits. Wu [12] achieved 98.15% AP and 0.965F1 using an improved EfficientNet-YOLOV4 dataset augmented by foliage occlusion data. However, its model capacity is 158 M, and the real-time detection speed is only 2.95 FPS. Chu [13] designed a novel Mask-RCNN for apple detection. By adding a suppression branch to the standard Mask-RCNN to suppress nonapple features, its F1 index is 0.905, but the detection speed is only 4 FPS. The suppression branch of this method is designed according to color, which is only effective when the color difference between fruit and leaf is large. When the color difference is not large, due to light, disease, or debris, the detection effect may not be good.

Although the above studies have all achieved some results for apple recognition in different scenarios, they all have similar problems. That is, high detection speed and high detection accuracy cannot be satisfied simultaneously. At the same time, according to the current research literature, several directions have been little studied. First, most of the current research on apple recognition has focused on apples that are dense, overlapping, or obscured by foliage, with very little research on apples in the context of bagging. Secondly, there are few studies related to apple detection models running on edge devices to determine how the detection models will perform in practice. To solve the above problems, an apple detection algorithm based on YOLOX-Tiny is proposed in this paper. It can meet the needs of a picking robot working with high precision and in real time. Compared to similar studies, our main contributions are the following two.

(1) A novel lightweight apple detector was designed. The ShufflenetV2-YOLOX model was designed from a practical perspective based on the orchard environment and obtained excellent detection speed and detection accuracy.
(2) It was validated and deployed on the Jetson Nano, an edge device. It was validated that the model can meet the requirements for real-time and high-precision detection on an edge device, and can provide an effective solution for picking robots.

## 2. Materials and Methods

### 2.1. Apple Image Acquisition and Data Augmentation

This paper takes the Fuji apple, the largest main apple variety in China, as the research object, and collects apple images from the apple demonstration base in Feng County, Xuzhou City, Jiangsu Province, China. Considering the possible natural environment in the actual orchard picking, the images of unbagged apples, bagged apples, and apples under weak light at night are collected.

In the process of image acquisition, in order to ensure the clarity of the image and meet the working environment of the picking robot, we keep the distance between the camera and the fruit at 0.3 m–2 m. In the night apple image acquisition, a single LED lamp is

used for illumination, and the brightness of the fruit area is changed by changing different illumination angles. A total of 1793 pictures are taken during the shooting, including apple images under different natural conditions such as forward light, backlight, side light, overlap, and occlusion, 577 apple images without bagging during the day, 567 apple images bagged during the day, and 649 apple images including bagging at night, as shown in Figure 1. Among them, the appearance of apples in the daytime will vary greatly due to the different angles and intensity of light. Bagging can not only prevent the fruit from being harmed by dust, pests, and pesticide residues, but also make the fruit surface smooth and beautiful, and increase the effective yield and income. However, due to a layer of plastic bags on the surface, the apple will be in an irregular state, and its surface and shape characteristics will be disturbed. This makes traditional image detection methods, such as texture, color difference, and Hough Circles transformation, unable to effectively detect apples [8]. At the same time, there are often water droplets in the plastic bag, which will bring greater difficulties to image detection. Because the image of apples at night is presented under the irradiation of a strong light source, there may be significant contrast on the same picture. For example, the surface of apples directly illuminated by the light source will be strong and bright, resulting in the lack of surface feature information, while those not directly directed will be relatively dark and difficult to detect. Therefore, apple images in the above cases will interfere with image detection to a certain extent [13].

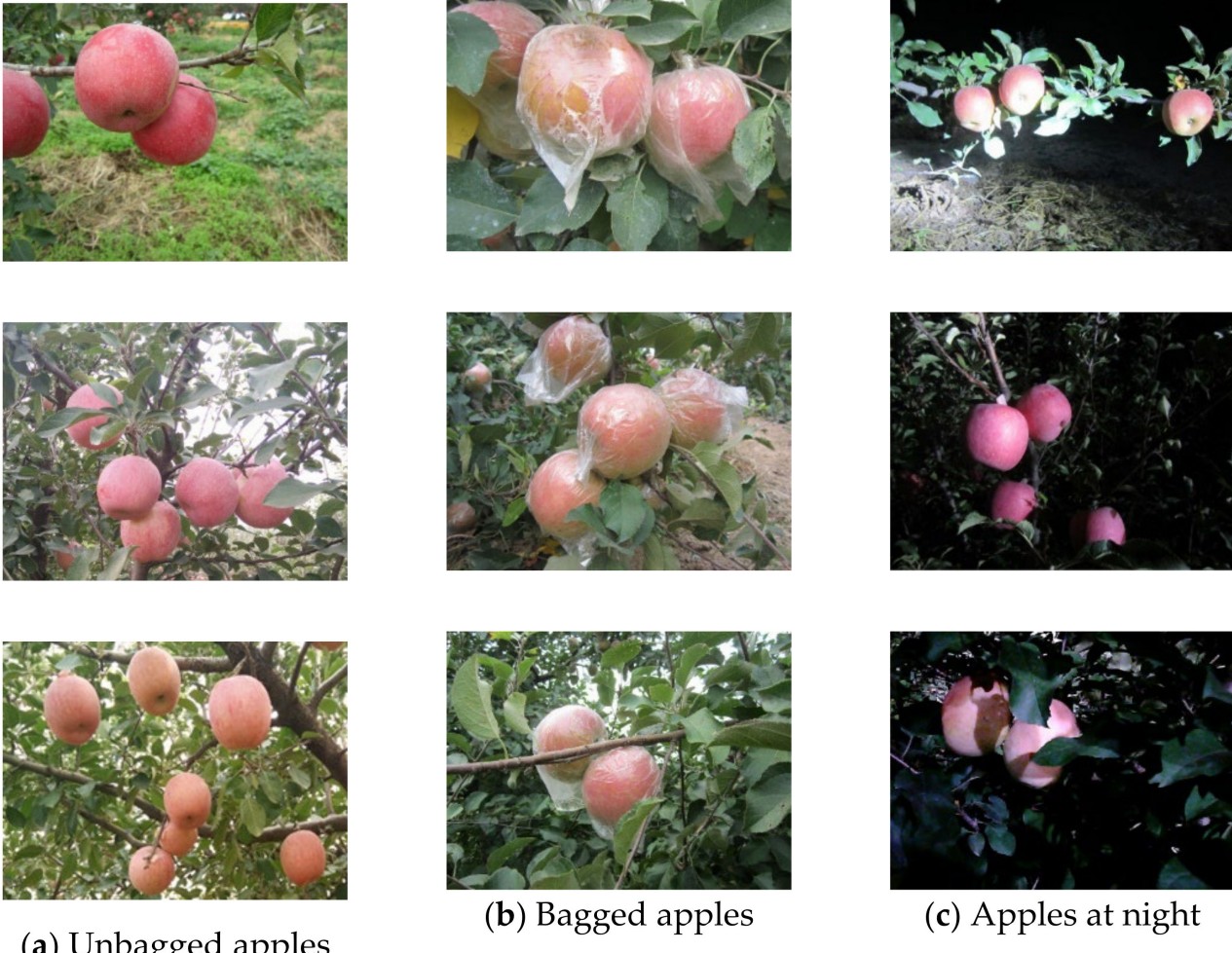

(**a**) Unbagged apples    (**b**) Bagged apples    (**c**) Apples at night

**Figure 1.** Apple image in natural state.

The apple dataset images collected in this experiment are small in number and contain complexities such as bagging, nighttime, occlusion, and overlap. Deep learning has certain

requirements on the size of the dataset. If the original dataset is relatively small, it cannot meet the training of the network model well, thus affecting the performance of the model. Image enhancement is the process of expanding the dataset by processing the original image, which can improve the performance of the model to a certain extent. Therefore, we use the imgaug algorithm for data enhancement, using mirror flip, changing brightness, flipping up and down, adding Gaussian noise, dropout, scaling, and other operations to mix and enhance the images with a 10-fold enhancement factor, while ensuring the morphological features are intact. Finally, 17,930 images are obtained, as shown in Figure 2. Although the augmented dataset is slightly different from the actual situation, the blurring is quite beneficial in improving the robustness of the model. The models trained with the data-enhanced dataset have higher accuracy compared to the unfuzzed dataset [14].

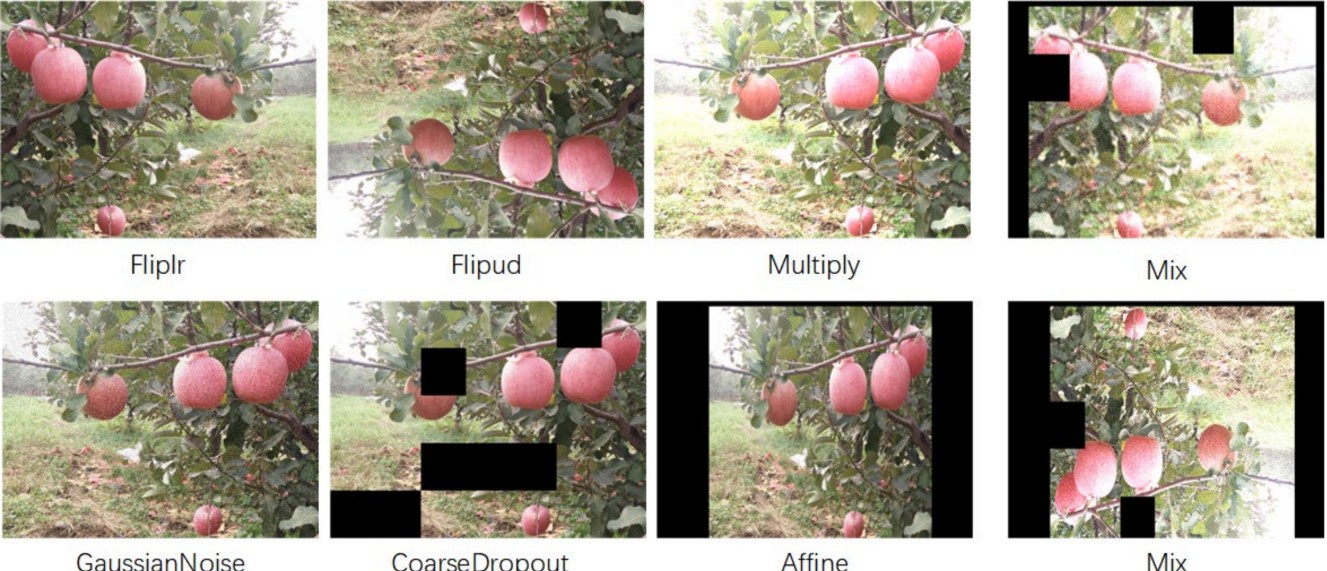

**Figure 2.** Image after data augmentation.

The annotation software used in this paper is LabelImg, and the annotation file format is "xml". To better compare different types of networks and training sets, the images are converted to Pascal VOC format. At the same time, the training set and verification set are generated according to the ratio of 9:1, and 30 apple images in the complex natural environment are selected as the test set to verify the detection effect of the model. All networks used in this paper are based on the pre-training of the ImageNet dataset, use migration learning to train 150 epochs on this dataset, and select the best one as the detection weight parameter to load into the network.

### 2.2. Design of Apple Object Detection Network

#### 2.2.1. Baseline Selection

There are a number of deep learning methods available, and one of the most effective networks for target detection is the convolutional neural network. These are divided into one-stage networks and two-stage networks [15–19]. The one-stage network is superior in detection speed, and the accuracy rate is also high. The neural network used for target detection is divided into the one-stage network and two-stage network according to the detection stage. The one-stage network is better in detection speed and high in accuracy. The YOLO series is a representative one-stage network, and among them, YOLOX is the latest version [20], which is improved with YOLOv3 + Darknet53 as the baseline. YOLOX adopts understanding coupling, Mosaic and Mixup image enhancement technology, anchor-free, SimOTA, and other tricks, which is greatly improved compared with the previous YOLOv4 and YOLOv5.

YOLOX is divided into x, l, m, s, tiny, and nano models from large to small according to the proportion of network depth and network width. Different models of networks can be selected according to different use scenarios. Among them, YOLOX-Tiny is a lightweight network in the YOLOX series, and its detection accuracy and speed are better than YOLOv4-Tiny, which is suitable for deployment in apple picking robots. However, the detection accuracy and detection speed of YOLOX-Tiny still have room for improvement compared with advanced apple detection algorithms at home and abroad.

### 2.2.2. ShufflenetV2-YOLOX Network Design

To meet the needs of the apple picking robot, it is necessary to improve the accuracy and detection speed of the network based on YOLOX-Tiny. This paper proposes a ShufflenetV2-YOLOX network. Figure 3 shows its network structure. First, this method takes YOLOX-Tiny as the baseline and uses the lightweight network Shufflenetv2 added with CBAM as the backbone. At the same time, ASFF is added after the PANnet network to improve the accuracy of network detection. Deleting a feature extraction layer reduces the amount of parameter calculation of the whole network, improves the detection speed of the network, and makes it meet the needs of real-time and high precision on embedded devices. The head network adopts YOLOX's decoupled head. It is divided into two parts: object prediction and position regression, which are predicted separately and then integrated for prediction. The loss function of the detection frame position can choose to use the traditional Intersection over Union (IOU) loss and Generalized Intersection over Union (GIOU) loss [21,22], and both OBJ loss and CLS loss use the Binary Cross Entropy loss method. To deal with the complex situation in orchard apple target detection, we select the better GIOU loss as the IOU loss of the detection frame.

$$IOU = \frac{S_{overlap}}{S_{nuion}} \tag{1}$$

$$GIOU = IOU - \frac{|A_C - S_{nuion}|}{A_C} \tag{2}$$

where $S_{overlap}$ is the area of intersection of the predicted bounding box and the true bounding box. $S_{union}$ is the area of the union of the two bounding boxes [14]. $A_c$ is the minimum enclosing rectangle that predicts the border and the true frame.

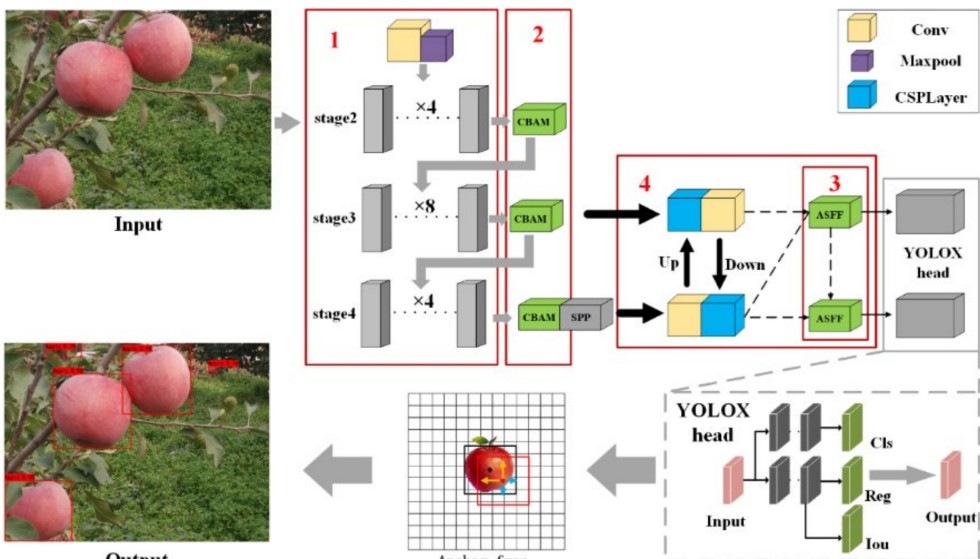

**Figure 3.** The network structure of ShufflenetV2-YOLOX. 1: Backbone Network Design; 2: Increase Attention Mechanism; 3: Add the ASFF Module; 4: Prune the Feature Layer.

Following the selection of the baseline model, the modeling phase is divided into four main stages. The first step is to replace the backbone network with ShufflenetV2. In the second step, the attention mechanism CBAM is added. The third step is to add the adaptive feature fusion mechanism ASFF. Finally, the feature extraction layer is reduced.

Backbone Network Design

YOLOX-Tiny is the lightweight network of YOLOX, which is achieved only by reducing the network width and depth. Compared with those specialized lightweight networks, it is not enough, so the first thing we need to do is to choose a lightweight network to replace YOLOX-Tiny backbone. ShuffleNetV2 is improved from ShuffleNet and has achieved excellent results in lightweight networks [23,24]. It inherits grouped convolution, depthwise separable convolution, and channel shuffle operations of ShuffleNet, and also improves the original unreasonable parts according to four efficient network pairs.

ShufflenetV2 is an image classification network in which the global average pooling and fully connected layers modules are added to achieve higher results in the ImageNet network competition and are useless for object detection networks. In order to replace the backbone of YOLOX-Tiny, we choose to keep only the network structure before stage4 in the ShufflenetV2 network, and then extract the output from each stage and connect it to PANet instead of CSPDarkNet. This can not only improve the running speed but also meet the design of the target detection network. The structure of ShufflenetV2 in YOLOX is shown in Figure 4.

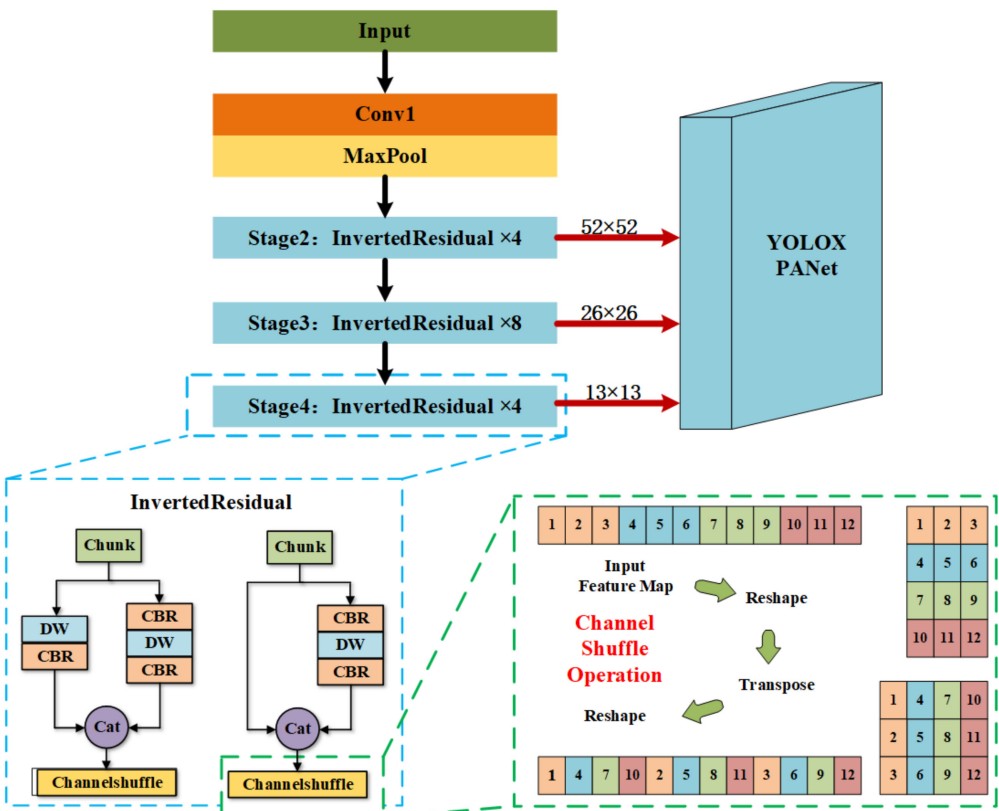

**Figure 4.** Shufflenetv2 network structure.

Increase Attention Mechanism

As the convolutional neural network (CNN) becomes deeper, the effective features become sparse. At this time, we need to introduce the "attention" mechanism. The attention mechanism can automatically learn and calculate the contribution of input data to output data so that it can ignore irrelevant noise information and focus on key information. CBAM is an attention mechanism module that combines space and channel [25]. Compared with

the SE attention mechanism that only focuses on channels, it can achieve better results. CBAM consists of a Channel Attention Module and Spatial Attention Module, which carry out Attention on the channel and space, respectively, as shown in Figure 5. In this paper, the CBAM module is added to the stage of the ShufflenetV2 backbone network, which can strengthen the apple features learned by the network.

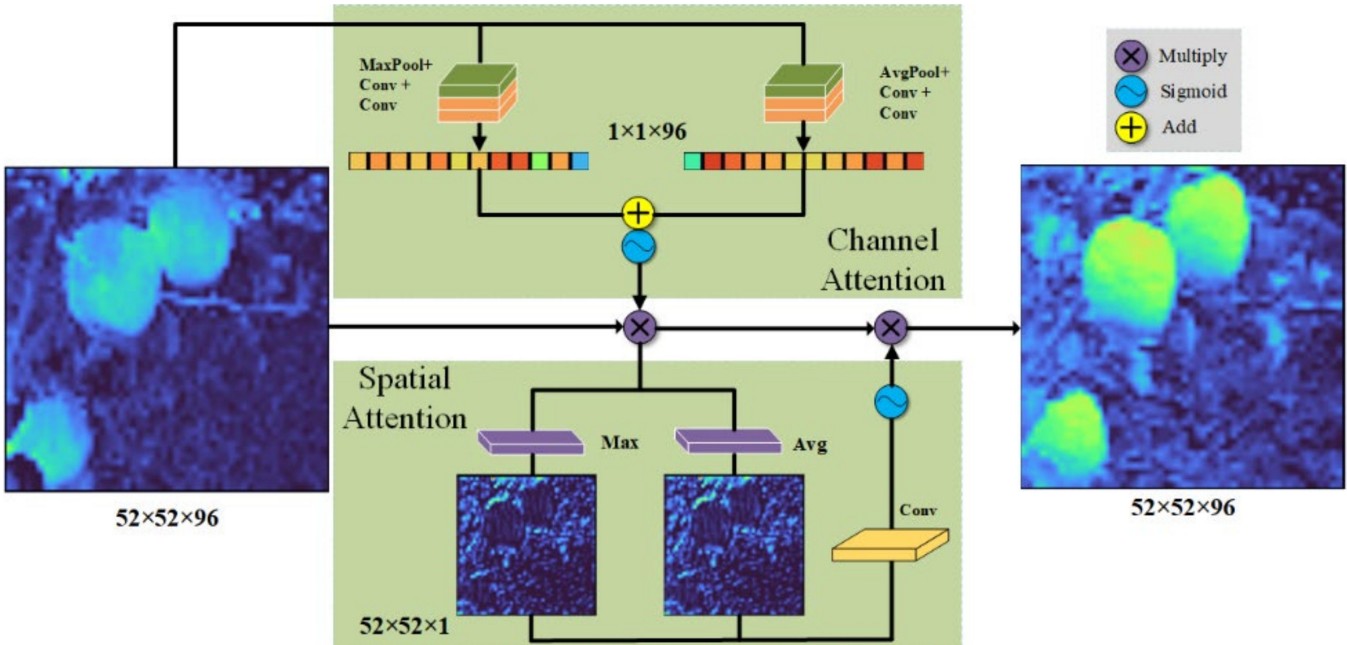

**Figure 5.** CBAM.

Add the ASFF Module

Feature pyramid can fuse features of different layers and detect images of different sizes, but the inconsistency between features of different scales is its main limitation. The ASFF module can make each feature layer focus on identifying objects that fit its grid size, spatially filter features on other layers, and retain only useful information for composition. This can solve the problem of indistinguishable fruits of different sizes clustered together in apple images [5]. Other layers in ASFF are adjusted to the same size as the current layer through convolution operations and fused to obtain adaptive weights. The adaptive weight is then combined with each layer to finally obtain a fusion module of the same size as the current layer. Its structure is shown in Figure 6. This paper adds an ASFF module after the PANet network to learn the relationship between different feature maps. This allows apples of different sizes to be predicted by the corresponding feature layers, improving the detection accuracy of the network.

Prune the Feature Layer

Adding modules can improve the detection accuracy of the network but also reduce the detection speed of the network. To improve the detection speed of the network to meet the real-time requirements, this paper chooses to delete one feature extraction layer in PANet and adjust the structure, and only uses two feature layers (TFL) to reduce the amount of calculation. Figure 7 shows the PANnet part of the YOLOX-tiny network. The black box shows the reduced network structure and the number of anchors. We only keep the 13 × 13 and 26 × 26 outputs, i.e., out2 and out3, respectively.

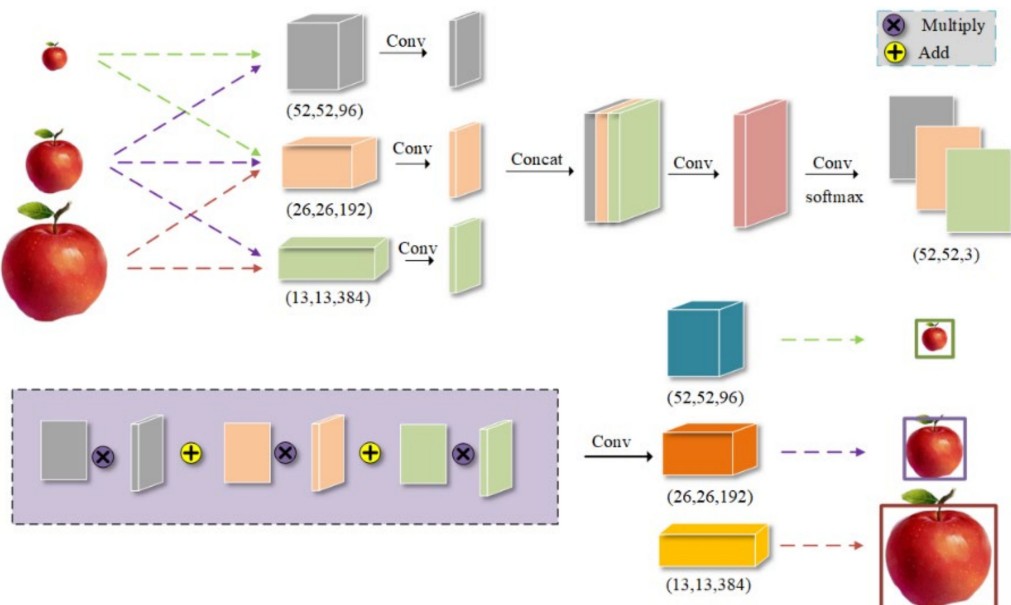

**Figure 6.** ASFF.

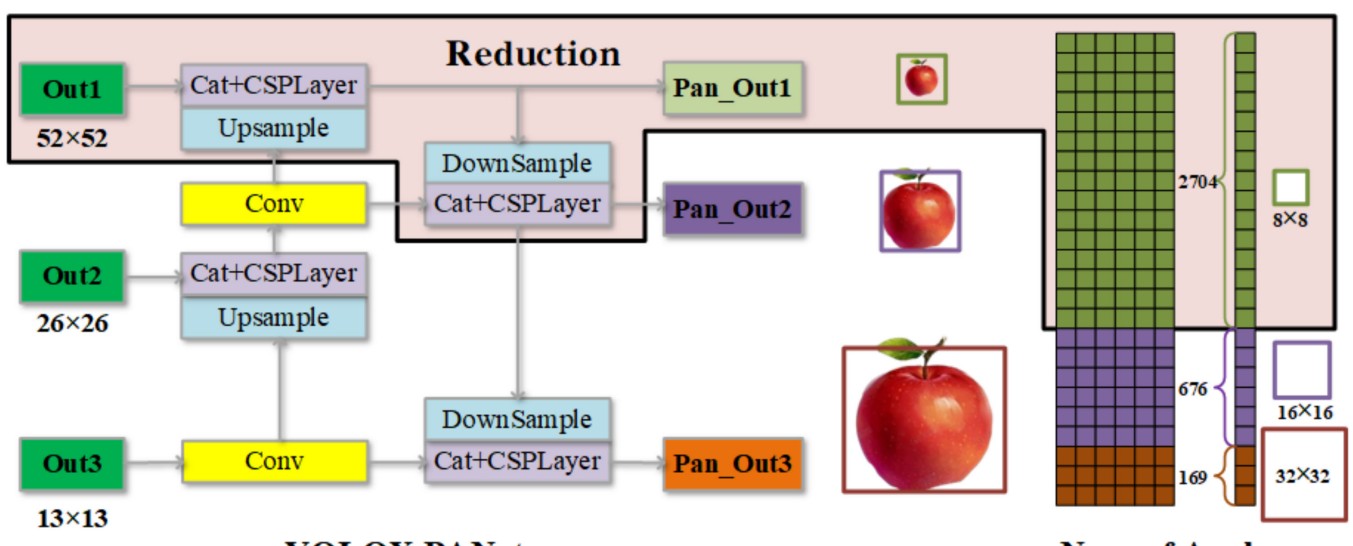

**Figure 7.** PANet with two feature layers.

Using only two feature layers can not only reduce the number of convolution kernels and computational complexity of PANet but also reduce the computing power required for prediction. In the case of $416 \times 416$ input size and num_class = 1, deleting a layer of feature layer Head will reduce the original 3549 anchors to 845 anchors. Although this will reduce the detection ability of small targets, it will not select long-distance small targets as objects during the operation of the apple picking robot, and the actual effect verification will not reduce the detection effect of the model too much.

## 3. Results and Discussion

The training equipment used in this paper is a PC device with the Windows 10 operating system. The system is equipped with an Intel e5-2683 processor, 64 GB of memory, four NVIDIA GTX1080ti graphics cards, and 11 GB of video memory. The algorithm programs used in this paper are written in the Python language on PyCharm, and CUDA and cuDNN are used for network training acceleration. The training epoch is

set to 150, the batch size is set to 64, and the input image size is set to 416 × 416. Input size and detection speed are mutually exclusive quantities and a smaller image input size speeds up detection. Therefore, the input image size is set to 416 × 416 to improve the real-time performance of the model detection. For all network models to compare performance fairly, the same input size needs to be set in the comparison experiments. This has a significant impact on the performance of the network models. The test equipment uses the Windows 10 operating system, AMD Ryzen7 4800 h processor, 16 GB of memory, an NVIDIA GTX1650 graphics card, and 4 GB of graphics memory (Table 1).

**Table 1.** Test System Hardware.

| Computer Configuration | Specific Parameters |
|---|---|
| Operating system | Windows 10 |
| CPU | AMD Ryzen 7 4800H |
| Random Access Memory | 16 GB |
| GPU | NVIDIA GTX1650 |

To verify the detection effect of the model on apples in the natural environment, this paper uses 30 complex orchard pictures as the test set, including 5 daytime unbagged apple pictures, 12 daytime bagged apple pictures, 10 nighttime unbagged apple pictures, and 3 nighttime bagged apple pictures. As nighttime and bagging are the focus and difficulty of the current research on picking robot vision, this paper chooses the nighttime and bagging test images to account for a higher proportion, which can better reflect the model's detection effect on apples in the natural environment.

In this paper, AP, Precision, Recall, Param, FPS, and F1 are selected as the comparison standards for detection effects to determine the pros and cons of the model. Param represents the number of parameters the network contains, and FPS represents the number of pictures the model can detect in one second. Taking the IOU threshold of 0.5 as the standard, the AP value is the area under the Precision—Recall (PR) curve formed by Precision and Recall. F1 score can be regarded as a weighted average of model accuracy and recall, which takes into account both the accuracy and recall of the model.

### 3.1. ShufflenetV2-YOLOX Model Performance Verification

To validate the effectiveness of the network improvement method, we chose to conduct ablation experiments to evaluate each step. AP, Param, and FPS were chosen as the evaluation metrics. The results of the ablation experiment are shown in Table 2.

**Table 2.** Ablation experiment.

| YOLOX-Tiny | ShufflenetV2 | CBAM | ASFF | TFL | AP | Param(M) | FPS |
|---|---|---|---|---|---|---|---|
| ✓ | | | | | 90.52% | 5.03 | 55 |
| ✓ | ✓ | | | | 91.69% | 3.19 | 53 |
| ✓ | ✓ | ✓ | | | 94.16% | 3.61 | 52 |
| ✓ | ✓ | ✓ | ✓ | | 97.29% | 6.68 | 48 |
| ✓ | ✓ | ✓ | ✓ | ✓ | 96.76% | 5.40 | 65 |

It can be seen from the data in Table 2 that each step of improvement is an effective improvement, which effectively improves the detection speed or detection accuracy of the model. The AP value of the ShufflenetV2-YOLOX method is 96.76%, which is 6.24% higher than that of the original YOLOX-Tiny method. Although the Param is increased by 0.4 m, the detection speed is increased by 18% to 65 FPS. Both the CBAM module and ASFF module effectively improve the detection effect of the network, and the method of deleting the feature layer also improves the detection speed within the range of tolerable reductions in accuracy. Due to the use of depthwise separable convolution and channel

shuffle operations in ShufflenetV2, when CSPDarknet is replaced, although the amount of network parameters is reduced, the detection speed is not improved.

Different deployment devices are suitable for different network structures. For example, a PC can use a CPU or GPU for inference. Depthwise separable convolutions are more suitable for running on CPUs, and normal convolutions are more suitable for running on GPUs. Due to the depthwise convolution and channel shuffle operations used in ShufflenetV2, inference on a GPU is not the best choice. Using ShufflenetV2 as the backbone network can achieve 15.6 FPS on the Ryzen7 4800 h(CPU), while YOLOX-Tiny can only achieve 11.5 FPS. In practice, we can choose different network structures based on different deployment devices.

### 3.2. Apple Detection Effect in Natural State

Apple recognition in complex environments has always been a research challenge. In this experiment, to verify the recognition effect of the trained model for different fruit states, apples without bags, apples with bags, and apples at night from the test set are detected. Figure 8 shows the apple detection results in a natural environment using the ShufflenetV2-YOLOX model. According to the detection results, the model proposed in this paper achieves good recognition results in various situations and meets the accuracy requirements of the apple picking robot.

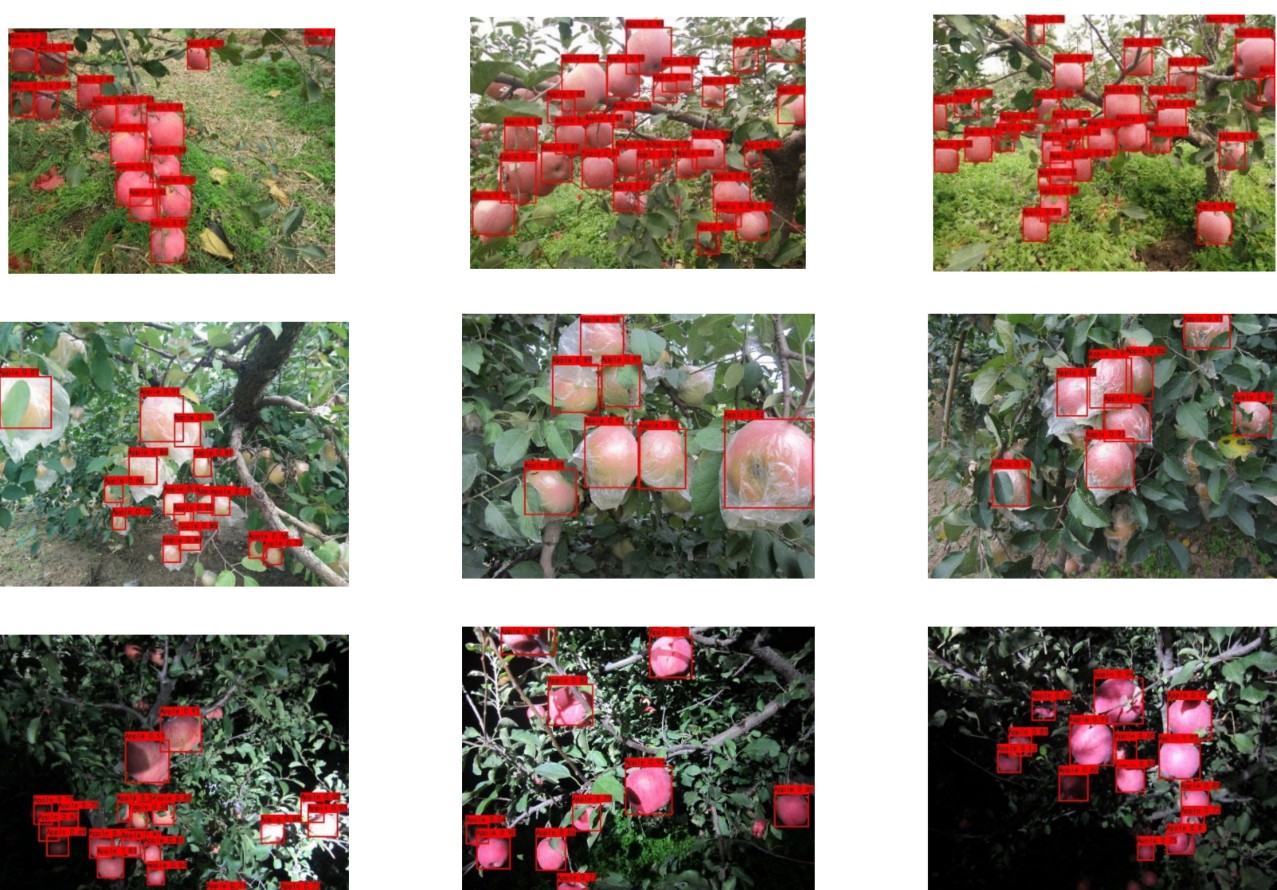

**Figure 8.** Apple detection effect in natural environment based on ShufflenetV2-YOLOX network model.

For images of unbagged apples during the day, the model can detect most of the apples, with only a few overlapping and too distant apples having detection errors. Images of bagged apples are not only sticky, overlapping, and obscured, but also irregular in shape due to the film on the surface of the bagged fruit. There are gaps between the fruit and

the film, which compromise the texture and color characteristics of the apple surface. As a result, bagged apples can be identified less accurately than nonbagged apples. Because of the low ambient light at night, apples close to a light source will have more distinctive features. As a result, apples close to the light source are easily detected, while apples away from the light source are difficult to detect. This is the biggest obstacle to nighttime image detection. In the future, the overall effect would be improved if more effort could be put into planning the lighting system to achieve more uniform illumination. Some obscured or small targets may not be detected, due to the limitation of the input image size of $416 \times 416$. All models have the same problem. Increasing the input size of the image can improve the detection of the model to some extent, but at the expense of detection speed. For example, ShuffleNetV2-YOLOX has a detection speed of 65 FPS at an input size of $416 \times 416$ and 60 FPS at an input size of $640 \times 640$. Although this is a reduction of 5 frames, the detection is much better and many small targets can be detected. However, the small targets are apple targets that are further away from the picking robot. For the apple picking robot, the small targets are not its working targets and do not affect the actual results. In subsequent work, a threshold pattern can be used, ignoring apples with a detection area smaller than a certain percentage. A target that focuses on a larger proportion of the area is an apple with a shorter distance. This facilitates the work of the picking robot.

Table 3 shows the precision and recall of the model detection in the three cases, the number of apples in the pictures, and the number of apples detected. There were 31 images containing 372 apple targets, of which 345 were detected. Our model can effectively address the low recall of apple detection networks under bagged and nighttime conditions.

**Table 3.** Detection results in different scenarios.

|  | Number of Images | AP | Recall | Number of Apples | Number of Apples Detected |
|---|---|---|---|---|---|
| Total | 31 | 96.76% | 93.75% | 372 | 345 |
| Unbagged apples | 11 | 97.29% | 94.45% | 134 | 125 |
| Bagged apples | 9 | 95.53% | 93.15% | 110 | 102 |
| Apple at night | 11 | 95.86% | 93.45% | 128 | 118 |

### 3.3. Apple Detection Effects Contrast Experiment of Different Models

To verify the superiority of the ShuffleNetV2-YOLOX model proposed in this paper, it is compared with YOLOv5-s, YOLOv4-Tiny, Efficientdet-d0, Mobilenetv2-YOLOv4-lite, and YOLOX-Tiny [7,26,27]. Figure 9 shows the apple detection results of ShuffleNetV2-YOLOX and other models in the natural environment. ShuffleNetV2-YOLOX, YOLOv4-Tiny, YOLOX-Tiny, Mobilenetv2-YOLOv4-lite, and YOLOv5-s have an image input size of $416 \times 416$, and Efficientdet-d0 has better results because its network settings have a fixed input size of $512 \times 512$. To make each model have a clearer contrast effect, this paper selects the apples detected by all models as the total set and marks the detection effect diagram of each model. The white circle indicates the missed area, and the blue circle indicates the missed area. The more white and blue circles, the worse the effect of the model. As can be seen from Figure 9, apple targets during the day are bright in color and distinct in shape. Most models perform best on unbagged apples during the day. On the other hand, the plastic bags on the surface of the apples can blur their color and shape characteristics, resulting in the target and background being too close together. Bagged apples are therefore very susceptible to missed detection. At night, apples under strong light and low light are difficult to detect due to illumination problems. However, the ShuffleNetV2-YOLOX model proposed in this paper has the least white and blue circles in the detection images, indicating that it has the highest recall rate. In particular, apple images in bagging and at night, although not all targets in the image are detected, have a significant advantage over other lightweight networks. This shows that the model can effectively solve the problem of low recall rate of the apple detection network under bagging and night conditions.

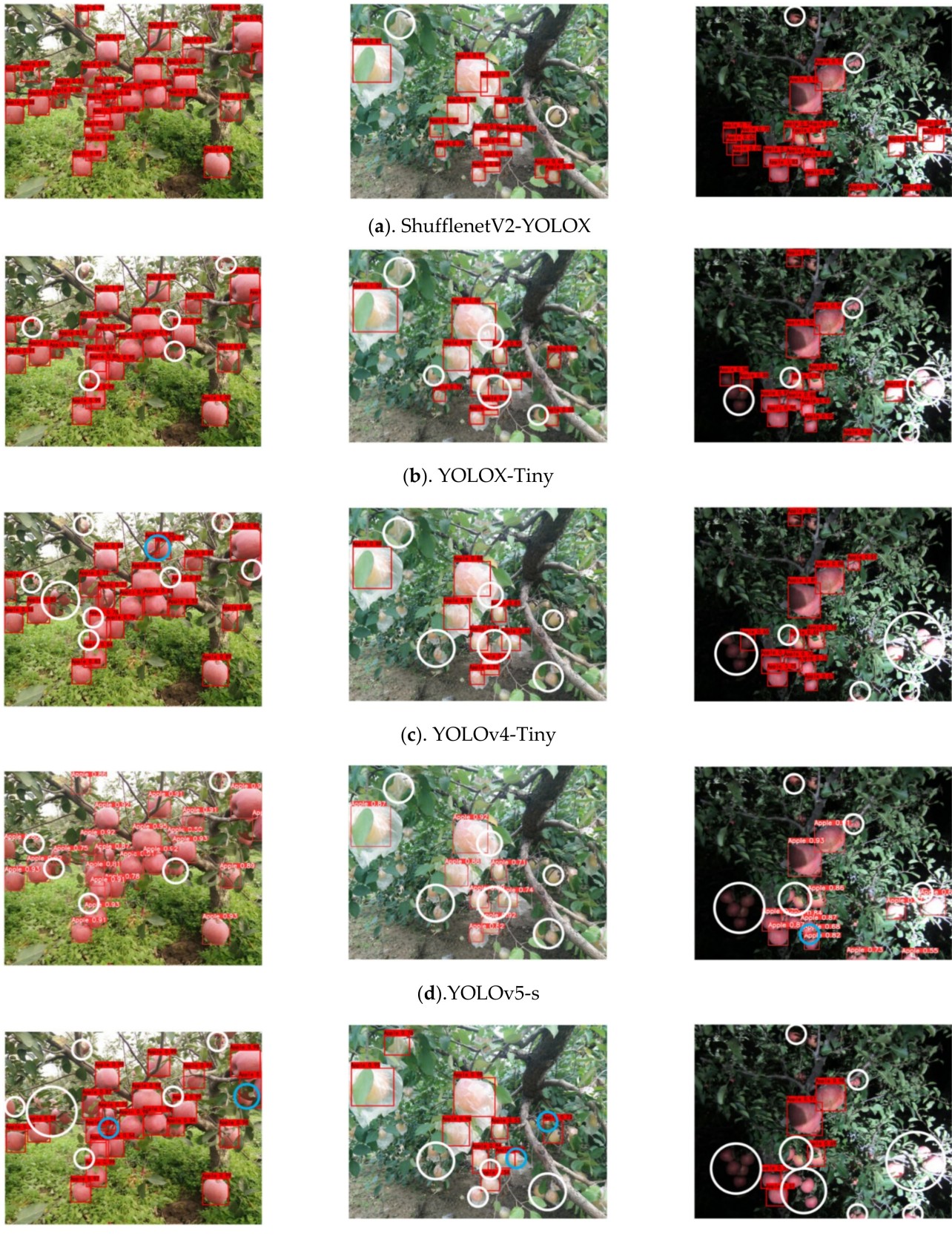

(**a**). ShufflenetV2-YOLOX

(**b**). YOLOX-Tiny

(**c**). YOLOv4-Tiny

(**d**).YOLOv5-s

(**e**). MobilenetV2-YOLOv4-lite

**Figure 9.** *Cont.*

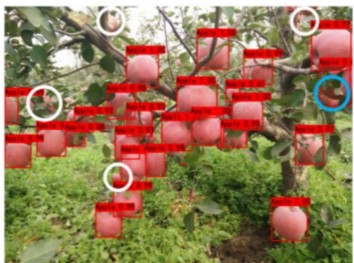 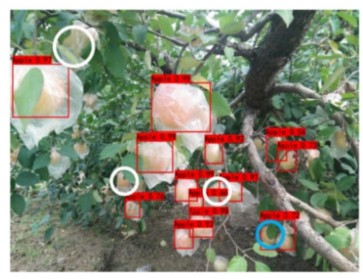 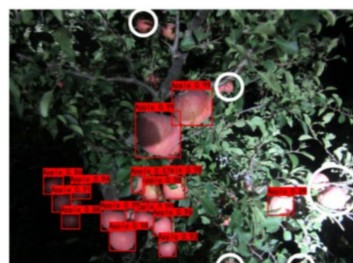

(**f**). Efficientdet-d0

**Figure 9.** Comparison of ShufflenetV2-YOLOX with other advanced networks for apple detection effects.

Figure 10 shows a comparison of the PR curves of the different models for apple detection. Table 4 shows a comparison of AP, precision, recall, F1, parameters, and FPS for the different models. In terms of detection accuracy, YOLOv4-Tiny is a simplified lightweight network from YOLOv4 with an AP of 89.14%, which is close to the performance of YOLOX-Tiny. YOLOv5-s is currently one of the best detection results among lightweight networks, with a relatively high recall and detection accuracy. The AP and F1 reach 95.44% and 0.94, respectively. Mobilenet-YOLOv4-lite achieves an AP of 92.99%. It has the highest accuracy of the tested models with 95.96%, but it does not have a high recall of 83.59%, which does not meet the apple target detection requirements. The performance of Efficientdet-d0 is similar to that of Mobilenet-YOLOv4-lite. The ShufflenetV2-YOLOX model proposed in this paper has a high recognition accuracy with an AP of 96.76% and a detection accuracy of 95.62%. In particular, the recall rate is the highest score among all lightweight networks, reaching 93.75%. Compared to other models, our model can effectively detect bagged and nighttime apple targets from low-resolution images, which is responsible for its high recall rate.

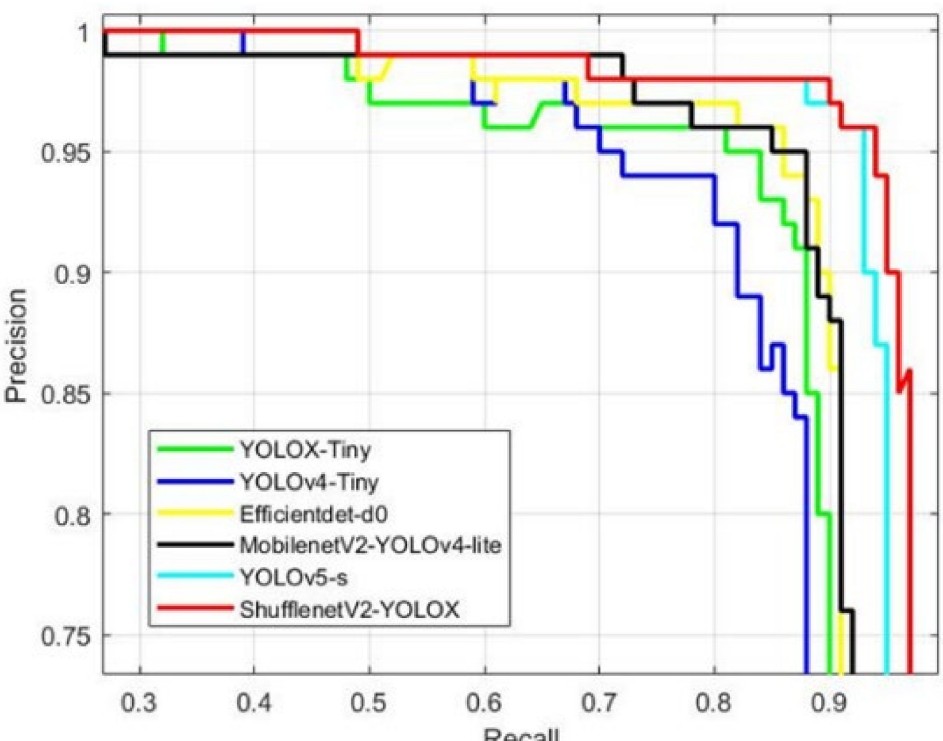

**Figure 10.** PR curve comparison of ShufflenetV2-YOLOX with other advanced networks.

**Table 4.** Comparison of ShufflenetV2-YOLOX with other lightweight networks.

| Models | AP | Precision | Recall | F1 | Param (M) | FPS |
|---|---|---|---|---|---|---|
| YOLOX-Tiny | 90.52% | 94.06% | 74.22% | 0.83 | 5.03 | 55 |
| YOLOv4-Tiny | 89.14% | 89.64% | 87.89% | 0.89 | 5.77 | 54 |
| Mobilenet-YOLOv4-lite | 92.99% | 95.96% | 83.59% | 0.89 | 10.30 | 22 |
| YOLOv5-s | 95.44% | 94.82% | 92.97% | 0.94 | 7.20 | 18 |
| Efficientdet-d0 | 92.89% | 95.91% | 86.42% | 0.91 | 3.69 | 21 |
| Ours | 96.76% | 95.62% | 93.75% | 0.95 | 5.40 | 65 |

In terms of detection speed, Yolov4-tiny and YOLOX-Tiny have an advantage in detection speed due to their lightweight network structure design, which can reach around 55 FPS. YOLOv5-s is a little slower at 18 FPS, and Efficientdet-d0 has fewer network parameters but is slow because it uses a lot of deeply divisible volume integrals. Although its floating-point operations per second (FLOPS) are small, it spends more time on memory access costs, so the speed is not ideal at 21 FPS. MobilenetV2-YOLOv4-lite uses MobilenetV2 to replace the YOLOv4 backbone, but the PANet is still large, and it uses deep detachable convolution instead of partial convolution, so the detection speed is not ideal, only 22 FPS. Our ShufflenetV2-YOLOX benefits from a lightweight backbone network with a low number of parameters. The anchor-free and two feature extraction layers can in turn reduce parameters and computations while satisfying the actual apple orchard detection. This results in a fast recognition speed of up to 65 FPS.

With higher detection accuracy and speed, ShufflenetV2-YOLOX enables real-time, accurate, and fast recognition of apples in natural environments, making it more suitable for deployment in apple picking robots.

### 3.4. Apple Detection Effect in Embedded Devices

Traditional deep learning algorithms use an Industrial Personal Computer (IPC) as the deployment device, which is not suitable for real-time apple detection in the field, due to its weight and power limitations. The edge device has powerful arithmetic power, small size, light weight and low power consumption. It can locally perform arithmetic processing on the collected data and is a good choice to replace IPCs, and NVIDIA Jetson Nano is the most cost-effective edge device available [10].

The apple picking experimental platform with Jetson Nano as the controller is shown in Figure 11. It mainly consists of a moving part, a gripper, a visual recognition system, and a robot arm. When the apple picking robot starts the picking task, it will first detect and select an apple through the visual recognition system. Then, it sends the apple's position information to the control system, and the robot arm is driven to approach the apple. The gripper will be driven to the designated position to grab the apple and use the cutter to cut off the stalk.

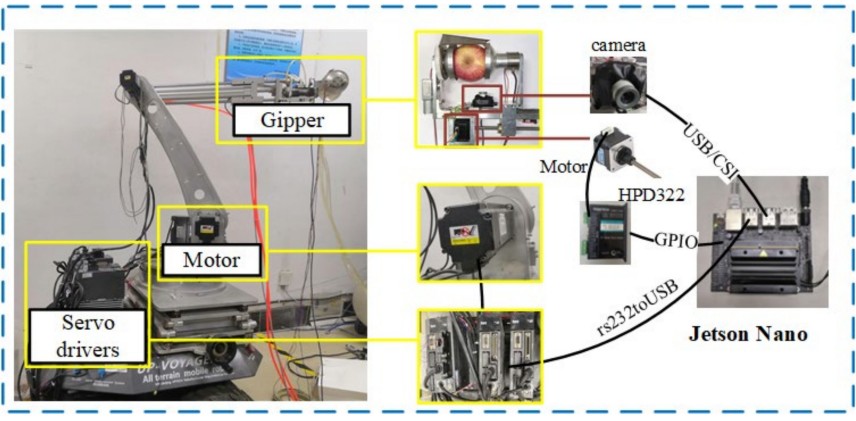

**Figure 11.** The apple picking experimental platform.

In this paper, we use Jetson Nano as an embedded deployment platform with software environment JetPack-4.5.1, TensorRT-7.1.3, and the image input size set to 416 × 416. The Pytorch model is first transformed into an ONNX model, and then TensorRT is used to quantify the accuracy of the parameters of the model and to merge the workflow so that it keeps the model running on the GPU as much as possible, thus allowing the model to run faster. We test the inference speed of the Pytorch Single-precision floating-point (FP32) model, ONNX INT64 model, TensorRT FP32 model, and TensorRT FP16 model on Jetson Nano. In Figure 12, the arrows refer to the increase or decrease in accuracy as a result of this operation compared to the previous phase.

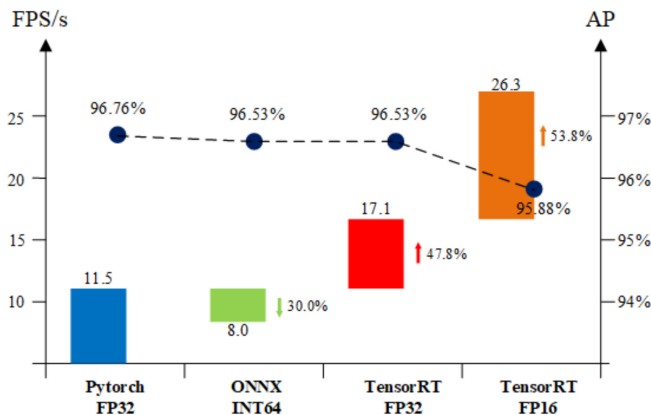

**Figure 12.** ShufflenetV2-YOLOX models for inference speed and AP accuracy on Jetson Nano.

On the Jetson Nano, the ShufflenetV2-YOLOX model with Pytorch FP32 can run at a speed of 11.5 FPS. The ONNX model, on the contrary, runs slower because of its parameter precision of double precision (INT64). As shown in Figure 11, we can see that TensorRT is very effective in accelerating the model. The TensorRT FP32 detects 47.8% faster with essentially no change in AP accuracy, reaching 17.1 FPS, while the TensorRT FP16 model detects 26.3 FPS with only a 0.88% loss in AP, a 53.8% improvement compared to the TensorRT FP32, and a 128.3% improvement compared to the original Pytorch FP32 model. ShufflenetV2-YOLOX is fully capable of meeting the real-time requirements of picking robots on embedded devices.

### 3.5. Comparison of ShufflenetV2-YOLOX with Existing Apple Target Recognition Methods

Table 5 gives the ShufflenetV2-YOLOX proposed in this paper as well as existing apple detection approaches. In the FPS column, PC and Edge indicate the speed at which the method runs in the computer and edge devices, respectively.

As can be seen from Table 5, the ShufflenetV2-YOLOX model proposed in this paper does not achieve the highest detection accuracy though, being 1.4 percentage points lower in AP compared to other methods mentioned in the literature. The possible reasons for this are considered: On the one hand, the dataset used in this thesis is complex, with three scenarios present. Each image contains an average of 12 apple targets, which raises the difficulty of apple detection. On the other hand, the network designed in this thesis is a light network, which focuses more on the operation speed of the network. Therefore, it is slightly lacking in detection accuracy. Compared with the methods in [12,13], the improved network in this thesis is more lightweight and improves the detection speed by 62 FPS and 61 FPS, respectively. The study in [10] can achieve a detection speed of 30 FPS on edge devices. However, the Jetson AGX Xaver it uses is eight times more expensive than the Jetson Nano used in this paper and is not cost-effective. Its AP is only 83.64%, well below our 96.76%.

Compared to the parameters in the literature, the ShufflenetV2-YOLOX model proposed in this paper has more outstanding advantages. Real-time detection can be achieved while ensuring detection accuracy.

**Table 5.** Comparison between ShufflenetV2-YOLOX and existing detection methods.

| Methods | Data Sets | Detection Network | Input Size | AP | FPS | F1 |
|---|---|---|---|---|---|---|
| Literature [10] | Dense apple | Improved YOLOv3-Tiny | 1920 × 1080 | 83.64% | 30 (Edge) | \ |
| Literature [12] | Apple shaded by leaves | EfficientNet-YOLOV4 | 416 × 416 | 98.15% | 2.95 (PC) | 0.96 |
| Literature [13] | Apple | Mask-RCNN | 280 × 720 | \ | 4 (PC) | 0.905 |
| Our | Unbagged apples, Bagged apples and Apple at night | ShufflenetV2-YOLOX | 416 × 416 | 96.76% | 65 (PC)/ 26.3 (Edge) | 0.95 |

## 4. Conclusions

To solve the problems associated with apple object detection in natural environments, this paper presented ShufflenetV2-YOLOX, an improved apple object detection method based on YOLOX-Tiny. The method was trained using a dataset of apples under daytime, bagged, and nighttime conditions. By replacing the backbone network, adding an attention mechanism, adding adaptive feature fusion, and reducing the number of feature extraction layers, the detection speed and detection accuracy of the model were improved.

The AP, accuracy, recall, F1, and FPS of the trained model were 96.76%, 95.62%, 93.75%, 0.95, and 65 FPS, respectively. A 6.24% improvement in AP and 10 FPS improvement in detection speed were achieved compared to the original YOLOX-Tiny network work. In addition, compared to the advanced lightweight networks YOLOv5-s, Efficientdet-d0, YOLOv4-Tiny, and Mobilenet-YOLOv4-Lite, the AP increased by 1.32%, 3.87%, 7.62%, and 3.77%, respectively, and the detection speed increased by 47 FPS, 44 FPS, 11 FPS, and 43 FPS, respectively. This shows that the feature fusion mechanism and the attention mechanism can improve the accuracy of apple detection in natural environments at an additional cost. The application of anchorless detectors overcame the drawbacks of past Anchor-based detectors, which were computationally intensive and reduced the setting of hyperparameters and post-processing. At the same time, the application of a lightweight backbone network and the use of only two feature extraction layers reduced the size of the model and increased the detection speed. For some embedded devices with low computational power, such as the NVIDIA Jetson Nano, the detection speed could reach 11.5 FPS, while with TensorRT acceleration, the inference speed of the TensorRT FP16 model reached 26.3 FPS at the expense of only 0.88% AP.

In summary, it offers significant advantages over other current lightweight networks in terms of detection speed and detection accuracy, and significantly improves recall rates for night and bagged apples. It can meet the requirements of real-time and high-precision detection for embedded devices. The method can provide an effective solution for vision systems for apple-picking robots.

**Author Contributions:** Conceptualization, W.J. and Y.P.; methodology, W.J. and Y.P.; software, Y.P. and J.W.; validation, J.W.; formal analysis, Y.P.; investigation, B.X.; data curation, Y.P.; resources, B.X.; writing—original draft preparation, Y.P.; writing—review and editing, W.J. and J.W.; visualization, B.X.; supervision, W.J.; project administration, W.J.; funding acquisition, W.J. and B.X. All authors have read and agreed to the published version of the manuscript.

**Funding:** This research was funded by the National Natural Science Foundation of China (No. 61973141), the Jiangsu agriculture science and technology innovation fund (No. CX(20)3059), and A Project Funded by the Priority Academic Program Development of Jiangsu Higher Education Institutions (No. PAPD-2018-87).

**Institutional Review Board Statement:** Not applicable.

**Data Availability Statement:** Not applicable.

**Conflicts of Interest:** The authors declare no conflict of interest.

**Abbreviations**

| | | | |
|---|---|---|---|
| ASFF | Adaptive Spatial Feature Fusion | AP | Average Precision |
| CBAM | Convolutional Block Attention Module | CNN | Convolutional Neural Network |
| FP32 | Single-precision Floating-point | FP16 | Half-precision Floating-point |
| FPS | Frames Per Second | FLOPS | Floating-point Operations Per Second |
| GIOU | Generalized Intersection over Union | IOU | Intersection over Union |
| INT64 | Double Precision | IPC | Industrial Personal Computer |
| PR | Precision-Recall | TFL | Two Feature Layers |

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
