# Peer review of "A Real-Time Apple Targets Detection Method for Picking Robot Based on ShufflenetV2-YOLOX"

_agriculture, doi:10.3390/agriculture12060856_

Round 1

Reviewer 1 Report

The manuscript is well-written and well-prepared as an original submission. It requires some major corrections as follows:

  • the research gap is lost in the introduction section.
  • please recheck the abbreviations in their first appearance. you can also prepare a table.
  • please implement the modeling phase description you can use the following original papers: "Mosavi, Amir, Sina Ardabili, and Annamaria R. Varkonyi-Koczy. "List of deep learning models." In International Conference on Global Research and Education, pp. 202-214. Springer, Cham, 2019.", "Tian, Yunong, Guodong Yang, Zhe Wang, Hao Wang, En Li, and Zize Liang. "Apple detection during different growth stages in orchards using the improved YOLO-V3 model." Computers and electronics in agriculture 157 (2019): 417-426." and other similar studies.
  • please indicate the innovation of the study in comparison with the similar studies.
  • please add a discussion section in comparison with the similar studies' findings to describe the reason of the findings.
  • please implement a proper conclusion.
  • How did you select the best model architecture for each scenario?

Author Response

Responses to Reviewer 1

We thank you for your comments and suggestions, which helped revise this paper. According to your suggestions, the following modifications have been made.

Question 1: the research gap is lost in the introduction section.

Response and modification: Thanks for your comment. We are sorry for not being able to express our views clearly, we have revised them. We have added the relevant content to the article. (Please see lines 74-79 in the revision paper.)

Question 2: please recheck the abbreviations in their first appearance. you can also prepare a table.

Response and modifications: Thank you for your comment. We are very sorry that our lack of care caused you trouble in reading it, and we have rechecked and completed all the abbreviations and created a table. (Please see line 487 in the revision paper.)

Question 3: please implement the modeling phase description you can use the following original papers: "Mosavi, Amir, Sina Ardabili, and Annamaria R. Varkonyi-Koczy. "List of deep learning models." In International Conference on Global Research and Education, pp. 202-214. Springer, Cham, 2019.", "Tian, Yunong, Guodong Yang, Zhe Wang, Hao Wang, En Li, and Zize Liang. "Apple detection during different growth stages in orchards using the improved YOLO-V3 model." Computers and electronics in agriculture 157 (2019): 417-426." and other similar studies.

Response and modifications: Thank you for your comments. The paper you provided was very helpful and we referenced and cited List of deep learning models and Apple detection during different growth stages in orchards using the improved YOLO-V3 model. A description of the relevant modelling stages has also been added in Chapter 2 as you suggested. (Please see lines 129-132, 145-147, 179-186, of the revised paper.)

Question 4: please indicate the innovation of the study in comparison with the similar studies.

Response and modification: Thanks for your comment. We are sorry for not being able to express our views clearly, we have revised them. There are two main innovations in our paper compared to studies of this type.

(1) A novel lightweight apple detector was designed. The ShufflenetV2-YOLOX model was designed from a practical perspective based on the orchard environment and obtained excellent detection speed and detection accuracy.

(2) Validated and deployed on the Jetson Nano, an edge device. Validation that the model can meet the requirements for real-time and high-precision detection on an edge device can provide an effective solution for picking robots. (Please see lines 81-88 in the revision paper.)

Question 5: please add a discussion section in comparison with the similar studies' findings to describe the reason of the findings.

Response and modifications: Thanks for your suggestion. According to your suggestion. We have added a comparison with the results of similar studies at the end as you suggested. (Please see lines 436-457 in the revision paper.)

Question 6: please implement a proper conclusion.

Response and modification: Thank you for your suggestion. We have made changes to the conclusion section. (Please see lines 468-477 in the revision paper.)

Question 7: How did you select the best model architecture for each scenario?

Response and modifications: Thank you for your comment and we are very sorry for the misunderstanding caused by the inappropriate expression. Firstly, we use the same model for apple detection in all scenarios, all modules are universal for improving model performance and the detection results in the scenarios are satisfactory to us. We have not chosen a model architecture for a particular scenario at this time. Moreover, the scenarios mentioned in our paper are at different times or different locations in the same orchard, and they belong to one whole. We will add a description of this in the experimental section. (Please see lines 337-342 in the revision paper.)

Reviewer 2 Report

A Real-Time Apple Targets Detection Method for Picking Ro-2 bot Based on ShufflenetV2-YOLOX

Journal: [Agriculture] Manuscript ID: agriculture-1715277

The vision-based apple-picking robot is designed that can detect and locate apples quickly and accurately in the orchard natural’s environment. The work is interesting and relevant to journal’s scope. The algorithm was developed with an adaptive spatial feature fusion module and image augmentation.

Overall, the presentation of the article is good. The topic is novel but there are also some issues.

Issues:

1. Theoretical analysis or discussion is not enough. Section 2 should analyze the theory further with reference literature- worked with image augmentation. For reference, some works were done for coral classification with deep learning and image augmentation. - Automated CNN Based Coral Reef Classification Using Image Augmentation and Deep Learning, 2021, International Journal of Engineering Intelligent Systems for Electrical Engineering and Communications 29(4):253-261.

2. Figure 7- caption should be brief and related.

3. What does it mean –“ the black box shows the parts to be deleted”?

4. Input image size fix – 416x416? Need some justification on pixel size.

5. Conclusion claim about speed in detection. It should have some numerical support and comparison with the various approach.

Author Response

Responses to Reviewer 2

We thank you for your comments and suggestions, which helped revise this paper. According to your suggestions, the following modifications have been made.

Question 1: Theoretical analysis or discussion is not enough. Section 2 should analyze the theory further with reference literature- worked with image augmentation. For reference, some works were done for coral classification with deep learning and image augmentation. - Automated CNN Based Coral Reef Classification Using Image Augmentation and Deep Learning, 2021, International Journal of Engineering Intelligent Systems for Electrical Engineering and Communications 29(4): 253-261.

Response and modification: Thanks for your comment. We are very sorry for the confusion in your reading. The main purpose of using data augmentation in this paper is to increase the number of datasets and improve the richness of the dataset. And to make the model more robust after training by adding some artificial interference. But this essay is equally informative. We have now added a theoretical analysis and discussion of image enhancement to the paper based on your suggestions. (Please see lines 120-124, 129-132 in the revision paper)

Question 2: Figure 7- caption should be brief and related.

Response and modifications: Thank you for your suggestion, we are very sorry for not explaining the picture succinctly, we have modified the title and picture in figure 7 to make it look short and clear. (Please see lines 241-245 in the revision paper.)

Question 3: What does it mean –“ the black box shows the parts to be deleted”.

Response and modifications: Thank you for your comment. We are very sorry for not being able to express our point clearly and we have revised it. Figure 7 shows the PANnet part of the YOLOX-tiny network. We have reduced the computation of the network and improved the detection speed of the network by deleting a layer of feature extraction. The black box shows the structure of the network we have deleted and the reduced amount of anchor computation. We have modified the caption and image in Figure 7. (Please see lines 241-245 in the revision paper)

Question 4: Input image size fix – 416x416? Need some justification on pixel size.

Response and modifications: Thank you for your comment. We are very sorry for not being able to express our views clearly. Input size and detection speed are mutually exclusive quantities and a smaller image input size speeds up detection. Therefore the input image size was set to 416x416 to improve the real-time performance of the model detection. For all network models to compare performance fairly, the same input size needs to be set in the comparison experiments. This has a significant impact on the performance of the network models. (Please see lines 259-264 in the revision paper.)

Question 5: Conclusion claim about speed in detection. It should have some numerical support and comparison with the various approach.

Response and modifications: Thank you for your comment. We are very sorry that we were unable to express our views clearly. We have now added a discussion on detection speed to the conclusion section as you suggested. (Please see lines 383-394 in the revision paper.)

Besides, section titles should be like "Introduction, Materials and Methods, Results, Discussion (can be combined with Results), Conclusions". Please include this in your revisions.

Response and revision: Thank you for your comment. We have now revised the section titles and structure of the paper in line with your suggestions.

Round 2

Reviewer 1 Report

All the comments have been successfully addressed.

Reviewer 2 Report

The paper is revised perfectly as per comments.